



**A new diagnostic for tropospheric ozone production**
Peter M. Edwards[1]* & Mathew J. Evans[1,2]
[1] Wolfson Atmospheric Chemistry Laboratories, Department of Chemistry, University of York,
Heslington, York, YO10 5DD, UK
[2] National Centre for Atmospheric Science, Department of Chemistry, University of York, Heslington,
York, YO10 5DD, UK
*pete.edwards@york.ac.uk
## Abstract
Tropospheric ozone is important for the Earth's climate and air quality. It is produced
during the oxidation of organics in the presence of nitrogen oxides. Due to the range
of organic species emitted and the chain like nature of their oxidation, this chemistry
is complex and understanding the role of different processes (emission, deposition,
chemistry) is difficult. We demonstrate a new methodology for diagnosing ozone
production based on the processing of bonds contained within emitted molecules, the
fate of which is determined by the conservation of spin of the bonding electrons.
Using this methodology to diagnose ozone production in the GEOS-Chem chemical
transport model, we demonstrate its advantages over the standard diagnostic. We
show that the number of bonds emitted, their chemistry and lifetime, and feedbacks on
OH are all important in determining the ozone production within the model and its
sensitivity to changes. This insight may allow future model-model comparisons to
better identify the root causes of model differences.
## 1. Introduction
The chemistry of the troposphere is one of oxidation [*Levy*, 1973; *Kroll et al.*, 2011].
Organic compounds together with nitrogen and sulfur containing molecules are
emitted into the troposphere where they are oxidised into compounds which can either
be: absorbed by the biosphere; are involatile enough to form aerosols; can deposit to
the surface; or be taken up by clouds and rained out. The oxidation of these
compounds is significantly slower than might be expected based on the atmospheric
composition of 20% molecular oxygen ($O_2$). The inefficiency of ground state $O_2$ as an
atmospheric oxidant is due to its electronic structure. With two unpaired electrons it
is a spin-triplet (total spin quantum number S=1, giving a term symbol of $^3\Sigma_g^-$). In



contrast, virtually all trace chemicals emitted into the atmosphere contain only paired
electrons and are thus spin-singlets (S=0). From a simplistic perspective (i.e. ignoring
nuclear spin interactions, inter-system crossings, nuclear dipole effects etc.) the spin
selection rule, $\Delta S=0$, means that the reaction of ground state $O_2$ with most emitted
compounds is effectively spin forbidden. Electronically excited $O_2$ ( $^1\Delta_g$ or $^1\Sigma_g^+$) is a
spin singlet and is more reactive in the atmosphere but low concentrations limit its
role [*Larson and Marley*, 1999]. Instead, atmospheric oxidation proceeds
predominantly via reactions with spin-doublet oxygen-derived species (S=½), notably
the hydroxyl (OH) and peroxy radicals ($RO_2 = HO_2$, $CH_3O_2$, $C_2H_5O_2$, etc.), or spin-
singlet species (e.g. ozone ($O_3$)).
One of the few spin-triplet species in the atmosphere other than $O_2$ is the ground state
of atomic oxygen ($O(^3P)$), which readily undergoes a spin allowed reaction with $O_2$ to
produce the spin-singlet $O_3$ molecule. This spin allowed reaction is responsible for the
creation of $O_3$ in both the stratosphere, where it forms the protective $O_3$ layer, and the
troposphere. The ability of $O_3$ to oxidise other spin-singlet species makes it a powerful
oxidant, and it is thus considered a pollutant with negative health effects. Sources of
$O(^3P)$ within the troposphere are limited because solar photons at sufficiently short
wavelengths to directly photolyse $O_2$ to $O(^3P)$ are essentially unavailable.
Aside from the photolysis of $O_3$ itself, the only other significant source of
tropospheric $O(^3P)$ is the photolysis of nitrogen dioxide ($NO_2$) [*Crutzen*, 1971].
Nitrogen oxides are emitted into the troposphere as nitrogen oxide (NO), which can be
oxidised to $NO_2$ by $O_3$ and other oxidants. A large thermodynamic energy barrier
prevents oxidation of NO to $NO_2$ by the OH radical [*Nguyen et al.*, 1998], and
therefore NO oxidation occurs through reaction with either $O_3$ or $RO_2$. In terms of $O_3$
production, the oxidation of NO by $O_3$ forms a null cycle. Thus only the reaction of
NO with $RO_2$ leads to a net production of $O_3$.
Exploring the distribution, source and sinks of tropospheric $O_3$ is a central theme of
atmospheric science. Chemical transport models (online and offline) are essential
tools enabling this understanding but their validity needs to be continually assessed.
Model-model comparison exercises are commonly performed to assess performance,
and comparisons of modelled $O_3$ budgets traditionally form part of this assessment
[*Stevenson et al.*, 2006; *Wu et al.*, 2007; *Wild*, 2007; *Young et al.*, 2013]. Ozone
production is diagnosed from the flux of NO to $NO_2$ via reaction with each of the



speciated $RO_2$ in the model's chemical schemes. This approach provides information
on the relative importance of the different $RO_2$ in the fast $NO + RO_2$ reactions within
the model, but gives very little detail on how the longer time scale model processes
(emissions, chemistry, deposition) influence $O_3$ production. Thus exploring the
reasons that models differ in their $O_3$ production is difficult and progress has been
slow.
A new diagnostic framework that links large scale model drivers such as emission,
chemistry, and deposition to $O_3$ production would allow an improved assessment of
why model ozone budgets differ.  We attempt to provide such a framework here.
## 2. A new diagnostic framework.
The rate of production of tropospheric $O_3$ is limited by the rate of oxidation of NO to
$NO_2$, which is in turn limited by the rate of production of peroxy radicals ($RO_2$).
Peroxy radicals form through association reactions of hydrogen (H) atoms or alkyl
radicals (both spin-doublets, $S=½$) with $O_2$, forming a highly reactive spin-doublet
radical on an oxygen atom. This spin allowed reaction converts spin-triplet $O_2$ that
cannot react with spin-singlet pollutants into a spin-doublet $O_2$ containing species that
can. As such the formation of $RO_2$ is central to the atmosphere's oxidation capacity,
and its production is limited by the rate of production of H atoms or alkyl radicals.
Thus the maximum potential rate of tropospheric $O_3$ production is equal to the rate at
which H atoms and alkyl radicals are produced.
Hydrogen atoms and alkyl radicals are predominantly produced via the spin allowed
breaking of the spin-pairing between the two electrons in a C or H containing covalent
bond ($S=0$), such as those in hydrocarbons. These spin-pairings can be broken in the
atmosphere either chemically or photolytically, with the products necessarily
conserving spin. The breaking of a covalent bond by a photon ($s=1$) can result in two
products with $S=½$ or two products with $S=0$. Likewise, oxidation by a radical ($S = ½$)
will result in one product with $S=0$ and one with $S=½$, because the unpaired electron
on the radical reactant pairs with one of the covalent bond electrons to produce a spin-
singlet.
Although the majority of $RO_2$ is formed from emitted C or H containing covalent
bonds, there are a few notable exceptions. Hydrogen atoms can also be produced
through the oxidation of CO to $CO_2$ by OH. During this reaction the coordinate bond



between the C and O atom is broken and the H atom is produced via the breaking of
the O-H bond. The other notable exception is the oxidation of an $SO_2$ lone pair of
electrons to $SO_3$ by OH, where again the H atom produced comes from the OH. In
both of these exceptions a spin-singlet electron pairing (CO coordinate bond or $SO_2$
lone pair) is broken during the production of the H atom, and we can therefore
consider these reactions as similar to the breaking of C or H containing covalent bond.
For simplicity these spin-singlet electron pairings that can be broken in the
troposphere to produce either a H atom or alkyl radical will be referred to as
"oxidisable bonds" (C-C, C-H, C=C, CO coordinate bond, S:).
Tropospheric $O_3$ production occurs through the oxidation of NO by $RO_2$. Following
the above rationale, these $RO_2$ are produced during the spin allowed breaking of
oxidisable bonds predominantly contained within emitted VOCs. This perspective
allows us to build a new metric for the production of tropospheric $O_3$ based around the
spin conserving properties of oxidisable bond breaking. In the extreme case, all
oxidisable bonds are photolysed to produce two spin-doublet $RO_2$ products, which
then react exclusively with NO to generate $O_3$. Thus at steady state, the maximum rate
of $O_3$ production is equal to the rate of production of $RO_2$, which is equal to twice the
rate of destruction of the number of oxidisable bonds. This in turn is equal to twice the
rate of emission of oxidisable bonds. Deviation from this maximum is determined by:
• The relative importance of processes that produce spin-singlet vs. spin-
doublet products during oxidisable bond breaking;
• The fraction of spin-doublet products from oxidisable bond breaking which
form $RO_2$;
• The fraction of $RO_2$ that go on to oxidize NO to $NO_2$.
To illustrate this Fig. 1 shows the tropospheric oxidation of a methane ($CH_4$) molecule
through various steps to either a carbon dioxide ($CO_2$) molecule or a species that is
deposited ($CH_3OOH$, $CH_2O$, $CH_3NO_3$). Methane contains 4 x C-H oxidisable bonds
(8 paired bonding-electrons) and as the oxidation proceeds, the number of oxidisable
bonds decays to zero. Figure 1 highlights the steps in the tropospheric $CH_4$ oxidation
mechanism that form spin-doublet products, with between 1 and 5 $RO_2$ produced
depending on the oxidation pathway. This compares with the theoretical maximum of
8 if all the original C-H bonds were photolysed to yield 2 spin-doublet products.



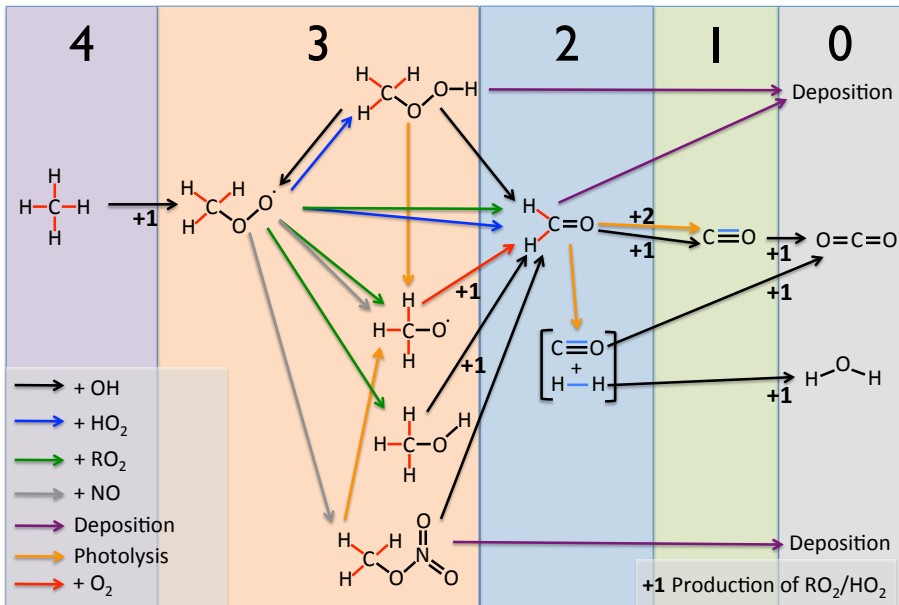


**Figure 1. Peroxy radical production during the tropospheric oxidation of CH₄.**
**Moving from left to right, the oxidisable bonds (emitted = red, produced = blue)**
**present in CH₄ are removed via a range of tropospheric processes, indicated by**
**the coloured arrows. The large numbers across top of the figure indicate the**
**number of oxidisable bonds at each stage of this oxidation. The production of**
**RO₂ is indicated by the +1/+2 numbers with the associated process arrows for**
**producing 1 or 2 RO₂ respectively.**
The principal atmospheric source of oxidisable bonds is the emission of C-H, C-C and
C=C bonds in hydrocarbons, with the only other significant sources being the
emission of CO and the chemical production of CO and $H_2$ during hydrocarbon
oxidation. Over a long enough timescale, the global atmosphere can be considered to
be in a chemical steady state, where the rate of loss of oxidisable bonds is balanced by
the rate of production or emission. Thus the $O_3$ production rate can be described by
equation (1), where the $O_3$ production metric $P_sO_3$ is equal to the number of spin-
paired electrons in oxidisable bonds (i.e. twice the sum of the number of oxidisable
bonds emitted ($E_{bonds}$) and chemically produced ($P_{bonds}$)), multiplied by the number of
spin-doublet radicals produced per oxidisable bond break divided by the maximum of
2 ($F_{Radicals}$), multiplied by the fraction of the radicals produced which are $RO_2$ ($F_{RO2}$),
multiplied by the fraction of $RO_2$ that goes on to react with an NO to produce an $O_3$





molecule ($F_{NO}$). A small correction (I) for the production of $RO_2$ via reactions of spin-
doublet radicals other than those that result in the breaking of oxidisable spin-pairings
(e.g. $O_3 + OH \rightarrow HO_2 + O_2$) is included.
$$P_sO_3 = \left( \left( 2 \times (E_{bonds} + P_{bonds}) \times F_{radicals} \times F_{RO_2} \right) + I \right) \times F_{NO} \qquad (1)$$

## 3. Implementation

We use the GEOS-Chem model to evaluate this new $O_3$ production diagnostic. GEOS-
Chem is a global chemical transport model of tropospheric chemistry, aerosol and
transport (www.geos-chem.org version 9-02). The model is forced by assimilated
meteorological and surface fields (GEOS-5) from NASA's Global Modelling and
Assimilation Office, and was run at 4°x5° spatial resolution. The model chemistry
scheme includes $O_X$, $HO_X$, $NO_X$, $BrO_X$ and VOC chemistry as described in Mao et al.
[2013] as are the emissions. The new $P_sO_3$ diagnostic has been implemented via the
tracking of reactions by type in the GEOS-Chem chemical mechanism file (further
details given in the SI). This tracking of reactions enables the fate of all oxidisable
bonds as well as the production and loss of all $RO_2$ within the model to be determined
using the standard GEOS-Chem production and loss diagnostic tools. Model
simulations were run for 2 years (July 1st 2005 – July 1st 2007) with the first year used
as a spin up and the diagnostics performed on the second year.
The standard GEOS-Chem diagnostic for $O_3$ production ($PO_3$) is shown on the left
side of Table 1. This emphasizes the very fast cycling between NO and $NO_2$, but
provides little in terms of higher process level information. The right side of Table 1
shows the new budget for $P_sO_3$, which tracks the processing of oxidisable bonds
within the model. Both diagnostic methods give the same final answer but our new
methodology provides more process level detail. Figure 2 illustrates this new process
based approach, showing the flow of emitted oxidisable spin-paired electrons (bonds)
to $O_3$ and the magnitude of the various mechanisms that contribute to and compete
with $O_3$ production. The annual oxidisable bond emission of 389 T mol yr[-1] has the
potential to create 778 T mol yr[-1] of radicals. If all oxidisable bonds were broken by
photons to produce two radical products the $RO_2$ production would be 778 T mol yr[-1].
If the oxidisable bonds were instead broken via radical reaction (e.g. OH) then $RO_2$
production would be 389 T mol yr[-1]. The various oxidisable bond breaking / removal



pathways within the model result in the production of 280 T mol yr$^{-1}$ of $RO_2$, with the
remainder largely producing stable spin singlet products.
Of the 280 T mol yr$^{-1}$ $RO_2$ produced, 112 T mol yr$^{-1}$ reacts with NO to produce $O_3$.
The remainder is lost through the reaction or deposition of $RO_2$ reservoir species
($RO_{2y}$= $RO_2$ + peroxides + peroxy-acetyl nitrates). For example the production of
methylperoxide ($CH_3O_2$ + $HO_2$ = $CH_3OOH$) results in the loss of 2 $RO_2$'s. However,
the reaction of methylperoxide with OH can re-release $CH_3O_2$ ($CH_3OOH$ + OH =
$CH_3O_2$ + $H_2O$). Thus, the production of methylperoxide represents the loss of a $HO_2$
and the movement of a $CH_3O_2$ into a peroxide $RO_{2y}$ reservoir species. The deposition
of a peroxide molecule is thus the loss of a $RO_{2y}$ reservoir species. Notable in Fig. 2 is
that the role of PAN and nitrate removal of global $RO_{2y}$ is negligible, instead being
dominated by peroxide production and loss and the reaction of $RO_2$ with $O_3$.

| $PO_3$ / T mol Yr$^{-1}$ | | $PO_3$ / T mol Yr$^{-1}$ (except $F_{Radicals}$, $F_{RO2}$, and $F_{NO}$ which are all unitless) | |
|---|---|---|---|
| NO + $HO_2$ → $NO_2$ | 74 | $E_{bonds}$ | 330 |
| NO + $CH_3O_2$ → $NO_2$ | 27 | $P_{bonds}$ | 58 |
| Other $RO_2$ + NO → $NO_2$ | 10 | $F_{radicals}$ | 0.40 |
| Other | 1 | $F_{RO2}$ | 0.86 |
| | | Inorganic $RO_2$ source | 15 |
| | | $F_{NO}$ | 0.40 |
| $PO_3$ | 112 | $P_sO_3$ | 112 |

**Table 1. Comparison of ozone production diagnostics for GEOS-Chem base**
**simulation. Standard model $PO_3$ diagnostics (left column) show reactions**
**responsible for NO to $NO_2$ conversions but provide little process level**
**information. The new $P_sO_3$ (right) provides increased information on the**
**processes controlling $O_3$ production within the model.**



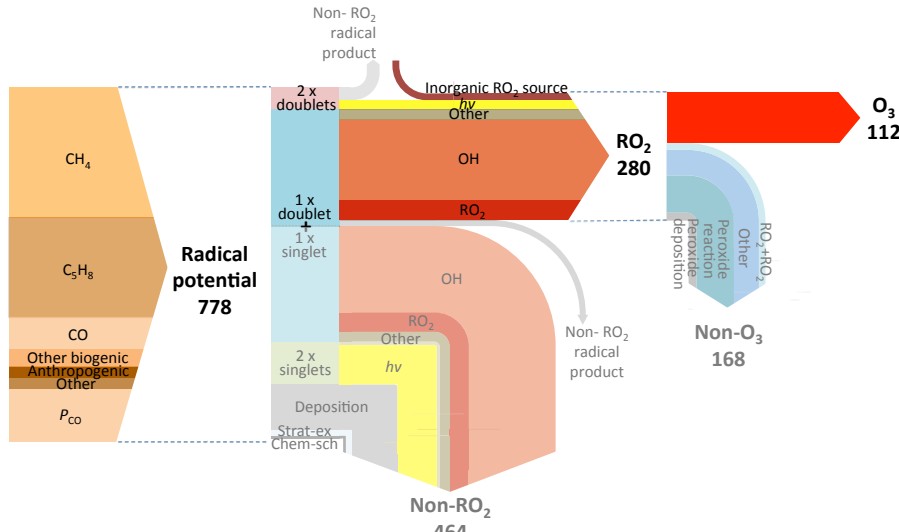

**Figure 2. Flow of oxidisable bonds to O₃ production in the GEOS-Chem base**
**simulation. Arrows are coloured according to process and the arrow thickness is**
**proportional to the flux through that channel. Spin-paired electrons are input as**
**oxidisable bonds into the model (left arrow), with the potential to create 778 T**
**mol yr⁻¹ of radicals. The actual fate of these bonds is shown in the central arrow,**
**producing 280 T mol yr⁻¹ of RO₂, of which 112 T mol yr⁻¹ reacts with NO to**
**produce O₃ (right arrow).**
## 3.1 Emitted oxidisable bonds
The fuel for tropospheric oxidation chemistry is the emission of oxidisable bonds,
predominantly in the form of hydrocarbons. The production of tropospheric $O_3$ from
the spin-paired bonding electrons emitted into the standard GEOS-Chem model
occurs with an efficiency of 14% (112 T mol yr⁻¹ molecules of $O_3$ produced / 778 T
mol yr⁻¹ spin-paired electrons emitted as oxidisable bonds, Fig.2). These spin-paired
bonding electrons are predominantly emitted in the form of $CH_4$, isoprene ($C_5H_8$) and
CO (37%, 28%, and 9% respectively). Oxidisable bonds produced during chemical
reactions ($P_{bonds}$), account for 15% of the net source. Figure 3 shows emissions of CO
and hydrocarbons in the standard GEOS-Chem simulation in terms of mass of carbon
per compound, number of oxidisable bonds per compound and as number of bonds in
different oxidisable bond types. The commonly used carbon mass approach splits
emissions approximately equally between each of the major sources ($CH_4$ (29%),



Isoprene (32%) and CO (30%)). In contrast, the oxidisable bonds accounting approach
apportions hydrocarbon emissions 44%, 33% and 11% for $CH_4$, isoprene and CO
respectively. This highlights the high number of oxidisable bonds per carbon atom in
$CH_4$ (4) compared to isoprene (2.8) and CO (1). Thus efforts to consider emissions on
a per-bond basis may provide more insight into chemical processes, as it is these
bonds that ultimately determine the chain-like chemistry rather than the mass of
carbon atoms. This helps to emphasise the relative importance of $CH_4$ emissions on
global tropospheric chemistry compared with other emissions such as isoprene or CO.
The type of oxidisable bond emitted is overwhelmingly C-H (71%).

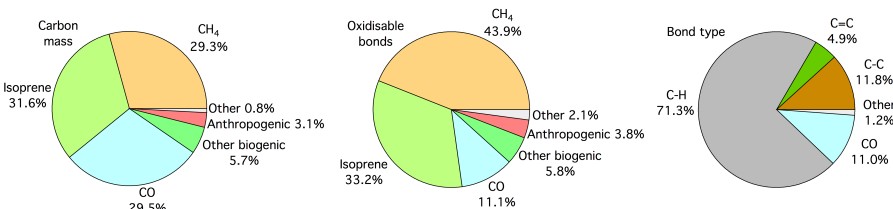


**Figure 3. Pie charts showing hydrocarbon emissions in the base GEOS-Chem**
**simulation. Emissions split by carbon mass (left), number of oxidisable bonds**
**(centre) and bond type (right).**
The total emission and production of oxidisable bonds has the potential to create 778
T mol yr$^{-1}$ of radicals. However, only 6% of the oxidisable spin-pairings are broken to
give the maximum 2 spin-doublet products (e.g. radical channel of $CH_2O$ photolysis).
The majority (68%) are oxidized via reaction with a spin-doublet species (OH) to
produce 1 spin-singlet and 1 spin-doublet product (e.g. OH + VOC). The remaining
26% of spin-paired electrons are removed to form two spin-singlets (e.g. the non-
radical channel of $CH_2O$ photolysis). Thus, of the 778 Tmol yr$^{-1}$ spin-paired electrons
emitted or produced only 265 T mol yr$^{-1}$ (34%) are converted into $RO_2$, with an
additional 15 T mol yr$^{-1}$ produced from reactions such as $O_3$ + OH → $HO_2$ + $O_2$ (*I*).
The efficiency of $O_3$ production from the available oxidisable bonds is further reduced
as only 40% of the 280 T mol yr$^{-1}$ of $RO_2$ produced react with NO to produce $NO_2$.
The remainder is lost either through the self-reaction of $RO_2$ or via loss through
deposition or reaction of $RO_{2y}$ reservoir species (e.g. peroxides). Thus overall 14% of
the emitted bonding electrons go on to make $O_3$.
This new $O_3$ production diagnostic shows the impact of processes such as emission,
deposition and chemical mechanism, providing significantly more detail than the
standard $PO_3$ diagnostic approach (Table 1). We now explore the sensitivity of model
$O_3$ production to changing emissions of $NO_x$ and VOC from the perspective of the
two diagnostic methods.

## 4 Model sensitivities

Understanding model response to changing emissions is an important tool for
considering policy interventions. The major controls on $O_3$ production are emissions
of $NO_x$ and VOCs. We show in Fig. 2 that from the perspective of global $O_3$
production, oxidisable bond emissions are dominated by $CH_4$ and isoprene. Figure 4
shows the impact of changing emissions of $NO_x$, isoprene and $CH_4$ on $O_3$ production
from both the perspective of this new methodology and the conventional $NO+RO_2$
diagnostic approach. The following sections investigate these model responses and
use the new diagnostic to provide insight into the processes driving the observed
response in $O_3$ production.

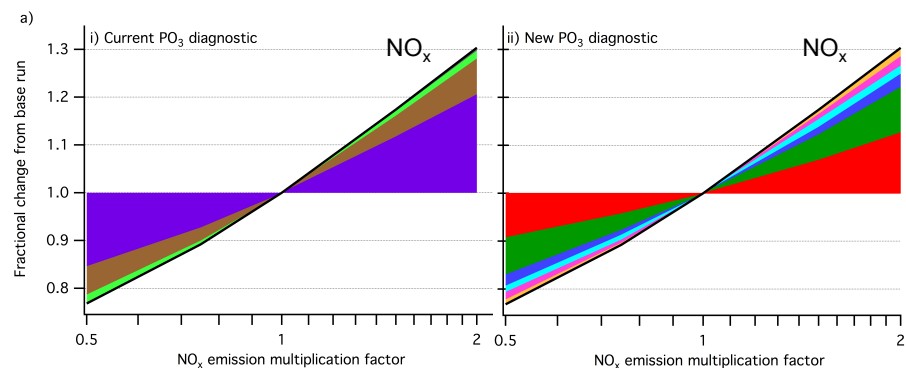


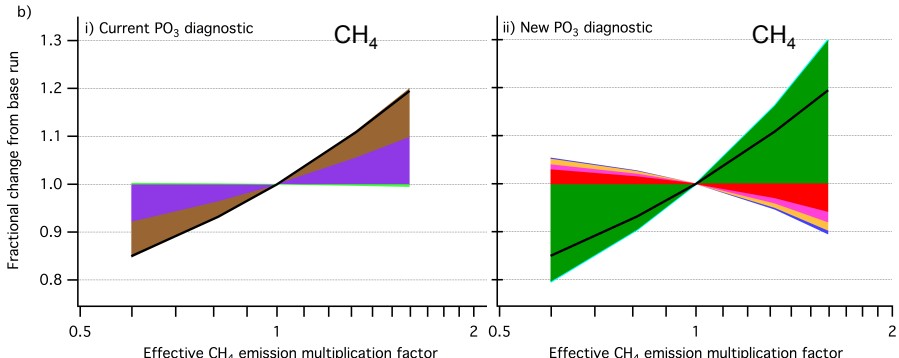




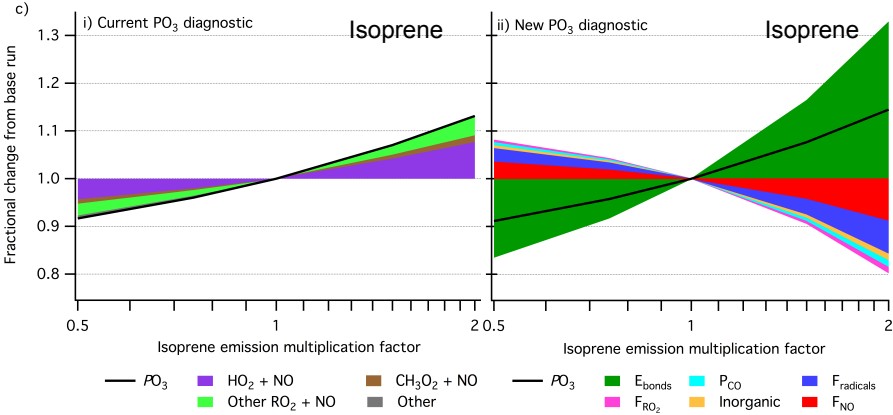


**Figure 4. Understanding the effect of NOx and VOC emissions on ozone production at the process level. Stack plots showing fractional change in model $PO_3$ compared to base simulation and associated contributions from the current $PO_3$ (i) and new $P_sO_3$ (ii) diagnostic parameters under changing $NO_x$ emissions (a), effective $CH_4$ emission (b) and isoprene emission (c).**

## 4.1 NOx emissions

Figure 4a diagnoses the relative response of GEOS-Chem $O_3$ production to changing $NO_x$ emissions, using simulations where $NO_x$ emissions from anthropogenic, biomass burning, biofuels, soil and lighting sources were multiplied by factors of 0.5 - 2. Increasing $NO_x$ emissions increases $O_3$ production. The standard $RO_2+NO$ diagnostic (Fig.4a(i)) shows that fractional contributions to the total change in $PO_3$ from $HO_2$ (67%), methyl-peroxy ($MO_2$) (25%), and other $RO_2$ (8%) remain approximately constant across the $NO_x$ emission range investigated. This diagnostic provides little detail on the processes driving the change in $O_3$ production under changing $NO_x$ emissions. In contrast, Fig. 4a(ii) is based on the new $P_sO_3$ diagnostic and shows a range of process level changes occurring as $NO_x$ emissions change.

### 4.1.1 Impact of changing NOx emission on $F_{NO}$

Unsurprisingly, as $NO_x$ emissions increase the fraction of $RO_2$ reacting with NO to produce $NO_2$ ($F_{NO}$) increases (red section in Fig. 4a(ii)). However, this impact only accounts for around 40% of the increase in $P_sO_3$. Figure 5a shows the fractional change in all the $P_sO_3$ efficiency parameters and the global mean $NO_x$ concentration as a function of the changing $NO_x$ emission. As $NO_x$ emissions increase the increase





in $NO_x$ concentration in the model is somewhat dampened. Halving the $NO_x$ emission
leads to $NO_x$ burdens dropping by ~35%, and doubling leads to an increase of 95%.
This dampening is due to the impact of $NO_x$ emissions on OH (see section 4.1.2),
which is the dominant sink for $NO_x$. Increasing $NO_x$ increases OH concentrations,
which in turn shortens the $NO_x$ lifetime thus dampening the response of concentration
to emission.

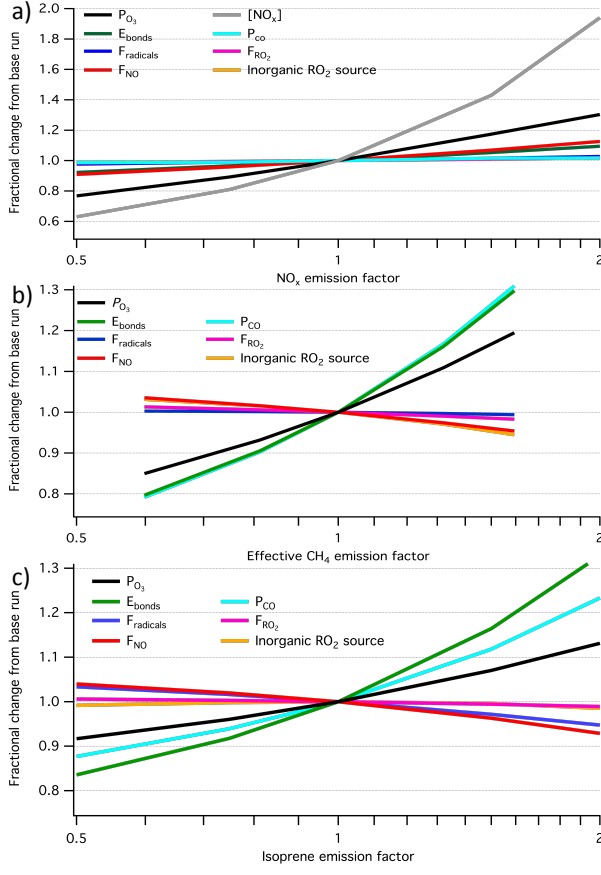


**Figure 5. Fractional change in new $P_sO_3$ diagnostic parameters from base run**
**against changing $NO_x$ emission (a); effective $CH_4$ emission (b); and isoprene**
**emission (c).**
The response of $F_{NO}$ to changes in $NO_x$ emissions is also dampened relative to the
change in $NO_x$ emissions. This is due to spatial variability in $F_{NO}$, which is not
affected uniformly by changing $NO_x$ emissions. Figure 6 shows the probability




distribution of $F_{NO}$ values across all model grid boxes for the base simulation and the
half and doubled $NO_x$ emission simulations (black, blue and red lines respectively).
For example, in a grid-box in the continental boundary layer where $RO_2$ reacts
overwhelmingly with NO, doubling the $NO_x$ emission may move $F_{NO}$ from 0.90 to
0.95 but it can't double it. Similarly, in the remote boundary layer where $RO_2$ reacts
overwhelmingly with other $RO_2$ doubling $NO_x$ emissions may move $F_{NO}$ from 0.3 to
0.4 but again it doesn't double. Thus the geographical spread of $NO_x$ chemistry limits
the change in $F_{NO}$ caused by changing $NO_x$ emissions. The spatial variability in the
new $P_sO_3$ diagnostic parameters shows that this approach has significant potential in
the analysis of regional $O_3$ budgets as well as global.

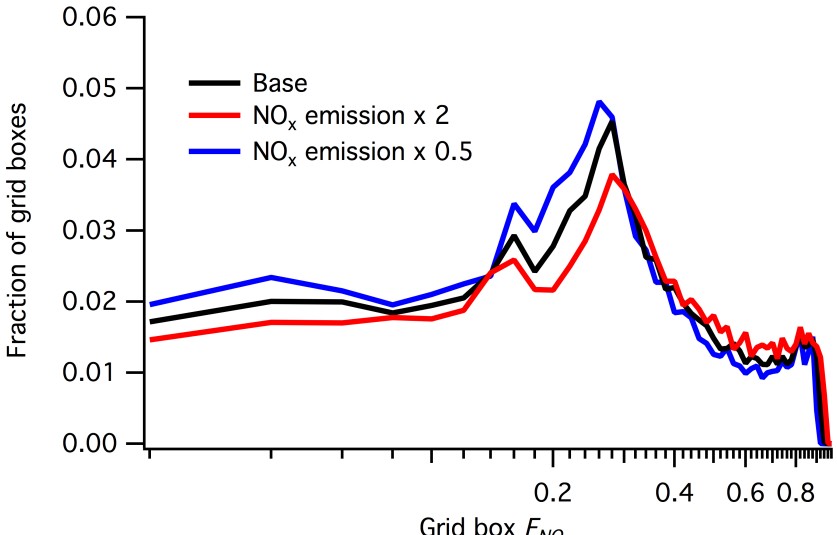


**Figure 6. Effect of $NO_x$ emission on distribution of $F_{NO}$ values (log scale). $F_{NO}$**
**values for each model grid box in the base and $NO_x$ emission x 0.5 and x 2**
**simulations, split into 50 x 0.02 width bins.**
**4.1.2 Impact of changing $NO_x$ emission on $E_{bonds}$**
Figure 4a(ii) shows that 60% of the response in $P_sO_3$ to changing $NO_x$ emission is due
to factors other than $F_{NO}$, with 40% of the increase due to changes in the emissions
($E_{bonds}$: 32%) and chemical production ($P_{bonds}$: 8%) of oxidizable bonds. This increase
in $E_{bonds}$ is surprising given VOC emissions are unchanged in these simulations.
However, increasing $NO_x$ emissions results in an increased OH concentration in the



model, which then leads to an increase in $CH_4$ oxidation. GEOS-Chem fixes $CH_4$
concentrations resulting in an increase in the effective $CH_4$ emissions as OH
concentrations increase, causing an increase in the total bond emission ($E_{bonds}$). Figure
7 shows the response of model global mean OH concentration and effective $CH_4$ bond
emission as a function of global mean $NO_x$ concentration across the simulations where
the base $NO_x$ emissions are multiplied by factors from 0.5 to 2. More $CH_4$ oxidation
also leads to more $CH_2O$ production and in turn more CO production ($P_{CO}$),
accounting for a significant fraction of the increase in this term.

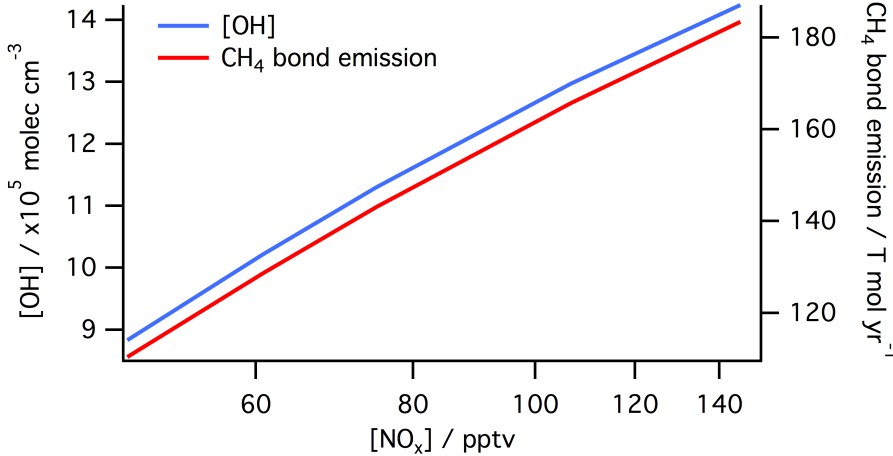


**Figure 7. Global mean OH concentration and effective $CH_4$ emission as a**
**function of [$NO_x$]. Plot shows effective $CH_4$ emission tracks OH concentration in**
**simulations where the $NO_x$ emission was increased or decreased from the base**
**simulation. Note X-axis log scale.**
**4.1.3 Impact of changing $NO_x$ emission on $F_{radicals}$, $F_{RO2}$ and $I$**
The fraction of radicals produced from bond oxidation ($F_{radicals}$) and the fraction of
those radicals which are $RO_2$ ($F_{RO2}$) show slight positive increase with $NO_x$ emission,
accounting for 9% and 6% of the change in $P_sO_3$ respectively. This reflects changes in
the partitioning of the fate of the oxidisable bonds, and is largely due to the changes in
OH.  As OH increases with $NO_x$ emission, the rate of chemical oxidation of bonds
increases at the expense of other losses, in particular deposition. The inorganic $RO_2$
source term ($I$) also correlates with $NO_x$ emission, as it is largely determined by the





concentrations of OH and $O_3$. This change accounts for 5% of the observed change in
$P_sO_3$.
Thus, with this new diagnostic methodology it is evident that only 40% of the model
$O_3$ production response to changing $NO_x$ emission is due to the direct effect of
increasing NO concentration on the rate of $RO_2 + NO$ reactions. Another 40% is due
to fixing the concentration of $CH_4$ within the model, with the final 20% due to the
increased OH competing for the available oxidisable bonds.
**4.2 Changing effective $CH_4$ emissions**
Figure 4b shows the effect on the $O_3$ production diagnostic of varying the prescribed
$CH_4$ concentrations by factors of between 0.5 and 2 from the base simulation. The
$CH_4$ emission rate (plotted) is diagnosed from the loss rate of $CH_4$ to reaction with
OH, the only $CH_4$ loss in the model. We describe this as the effective $CH_4$ emission.
As effective $CH_4$ emission increases, $O_3$ production also increases. The standard
diagnostic (Fig.4b(i)) shows that this increase occurs through an increased rate of
reaction of $HO_2$ and $CH_3O_2$ with NO, as would be expected as these are the $RO_2$
produced during $CH_4$ oxidation. The rate of other $RO_2 + NO$ reactions actually
decreases slightly as $CH_4$ emissions increase, due to lower OH concentrations and
increased competition for NO from $HO_2$ and $CH_3O_2$. The new diagnostic (Fig.4b(ii)),
however, shows the increase in $O_3$ production with increasing effective $CH_4$ emission
is not simply a result of more $HO_2$ and $MO_2$.
**4.2.1 Impact of changing effective $CH_4$ emission on $F_{NO}$**
The observed change in $P_sO_3$ is around one third smaller than would be expected from
the increase in the oxidisable bond emission ($E_{bonds}$) and bond production ($P_{bonds}$)
terms alone. This is due to a countering decrease in the other efficiency parameters
with increasing effective $CH_4$ emission. Figure 5b shows the fractional change in all
the efficiency parameters as a function of the changing effective $CH_4$ emission. The
decrease in the fraction of $RO_2$ reacting with NO to produce $NO_2$ ($F_{NO}$) is driven by
increasing $O_3$ concentrations, which push the $NO/NO_2$ ratio towards $NO_2$. This
reduces the availability of NO to react with $RO_2$ thereby reducing $O_3$ production. This
shift in the $NO/NO_2$ ratio also increases $NO_x$ loss within the model with increasing
$CH_4$ emission, as the increased $CH_4$ oxidation increases $RO_2$ concentrations resulting





in larger losses of $NO_2$ via compounds such as peroxyacetyl nitrate (PAN) and
peroxynitric acid (PNA).

**4.2.2 Impact of changing effective $CH_4$ emission on $E_{bonds}$**

As $CH_4$ is the largest single source of oxidisable bonds (Fig. 3), increasing the
effective $CH_4$ emission results in an increase in $E_{bonds}$. Changing the fraction of total
emitted oxidisable bonds from $CH_4$ does however have significant consequences on
the loss mechanisms of these bonds, which influences the other efficiency parameters.
The pie charts in Fig. 8 show the split of oxidisable bond loss mechanisms in the base
simulation and those with the $CH_4$ concentration fields multiplied by 0.5 and 2. As the
effective $CH_4$ emission increases the fraction of bonds lost via OH decreases, despite
the actual number of oxidisable bonds lost to OH increasing. A larger fraction of
bonds are therefore lost via the other mechanisms shown in Fig. 8 rather than reaction
with OH. As $CH_4$ removal occurs predominantly in the free troposphere, increasing
the effective $CH_4$ emission also results in a reduction in the fraction of oxidisable
bonds lost via deposition. The largest fractional increase in bond loss mechanism with
increasing effective $CH_4$ emission is for photolysis, with the increase in the "other"
fraction due to increased loss of bonds to the stratosphere with increasing $CH_4$. The
fraction of bonds lost via other chemistry (e.g. non-OH radical oxidation and $RO_2$ self
reactions) remains approximately constant across the effective $CH_4$ emission scenarios
investigated.

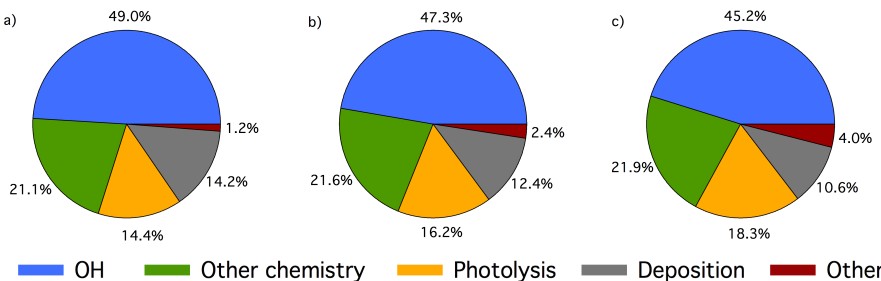

**Figure 8. Oxidisable bond loss mechanisms under changing $CH_4$ emissions. Pie charts showing fractional loss mechanisms for oxidisable bonds in model simulations with 0.5 x $CH_4$ concentration field (a), base simulation (b) and 2 x $CH_4$ concentration field.**





### 4.2.3 Impact of changing effective $CH_4$ emission on $F_{radicals}$, $F_{RO2}$ and $I$

The fraction of oxidisable bonds that goes on to produce radicals ($F_{radicals}$) and the fraction of these that are $RO_2$ ($F_{RO2}$) also decrease with increasing effective $CH_4$ emissions. This is due to decreasing global OH concentration resulting from increased loss by reaction with $CH_4$ and a decreasing NO concentration. This favours bond loss via pathways such as deposition rather than those that produce $RO_2$. These changes are predominantly due to the chemistry of $CH_2O$. As shown in Fig. 1, the oxidation of $CH_2O$ occurs either via reaction with OH or photolysis, with OH reaction yielding 1 $RO_2$ from the net breaking of 2 spin-singlet bonds, and the two photolysis channels yielding either 0 x $RO_2$ (spin-singlet products molecular channel) or 2 x $RO_2$ (spin-doublet products radical channel), with the molecular channel being dominant. The reduction in OH concentration with increasing $CH_4$ means photolysis increases its competition as a bond loss mechanism, which has the effect of reducing the average $RO_2$ production per $CH_2O$ oxidised. The increase in the fraction of total bonds lost through the $CH_2O$ photolysis as $CH_4$ increases thus results in a reduction in both $F_{radicals}$ and $F_{RO2}$. The reduction in $F_{radicals}$ due to changing $CH_2O$ fate, however, is largely offset by a reduction in the fraction of bonds lost via deposition as $CH_4$ increases. This is due to the long lifetime of $CH_4$ compared with the majority of other sources of oxidisable bonds, resulting in oxidation increasing fractionally in the free troposphere where deposition is a less significant loss mechanism than in the boundary layer.

### 4.3 Changing isoprene emission

The species through which the oxidisable bonds are emitted has a significant impact on $O_3$ production, due to their subsequent removal mechanisms. For example, in a simulation where the only emission of oxidisable bonds is CO, $F_{radicals}$ is 0.5 and $F_{RO2}$ is 1 as the only CO sink is reaction with OH to produce one $HO_2$ (OH + CO → $HO_2$ + $CO_2$). The CO coordinate bond, which in theory has the potential to produce 2 radicals, only produces 1 radical, which is an $RO_2$.

Isoprene has the most complex chemistry in the model and is the second largest source of bonds into the atmosphere (Fig. 3). Figure 4c shows the response of the two $O_3$ production diagnostics to varying the isoprene emission within the model. The standard diagnostic (Fig.4c(i)) shows that the most significant increase in $PO_3$ from





increasing isoprene emissions is from NO + $HO_2$ and non-$MO_2$ peroxy radicals, with a
smaller increase from $MO_2$. The new $P_sO_3$ diagnostic (Fig.4c(ii)) again provides more
insight, showing significant offsetting of around a half between the terms.

**4.3.1 Impact of changing isoprene emission on $F_{NO}$**

The increased isoprene emission leads to a similar change in the magnitude of the
total number of oxidisable bonds emitted ($E_{bonds}$) as the simulations in which effective
$CH_4$ emission were varied. However, the countering decrease in all of the efficiency
parameters is much larger for isoprene than for $CH_4$. Figure 5c shows the fractional
change in the new $P_sO_3$ ozone production diagnostic parameters as a function of
isoprene emissions compared to the base simulation. The change in $F_{NO}$ is due to both
a decrease in global mean $NO_x$ concentrations with increasing isoprene and the spatial
distribution of isoprene emissions. With the majority of global isoprene emissions
being in regions with low $NO_x$ emissions, and thus low values of $F_{NO}$. Figure 9 shows
a decrease in global mean $NO_x$, and global mean OH, concentrations with increasing
isoprene emissions, however, the effect is less than that seen when $CH_4$ is responsible
for the same increase in oxidisable bond emission. This is due in a large part to the
spatial scales over which the two compounds impact.

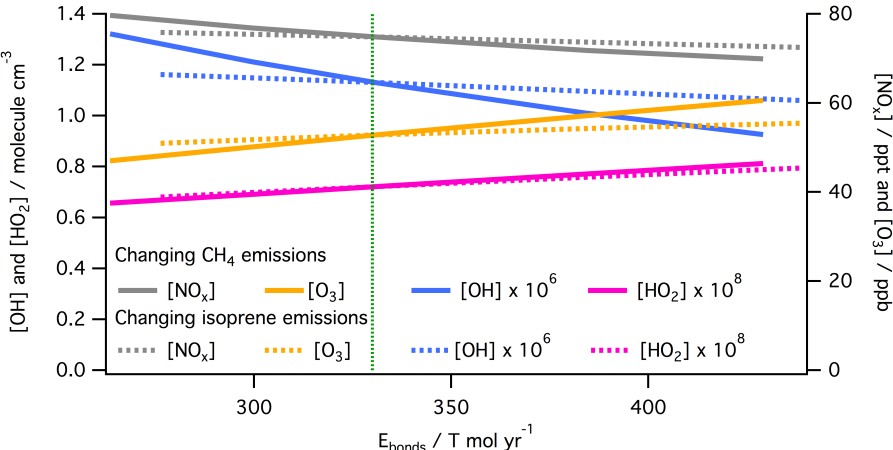


**Figure 9. The effect of oxidisable bond parent species on OH, $HO_2$, $O_3$ and $NO_x$ concentrations. Global mean [OH], [$HO_2$], [$O_3$] and [$NO_x$] for simulations where the effective $CH_4$ emission (solid lines) and isoprene emission (dashed lines) were changed, against model $E_{bonds}$. The dashed vertical green line indicates $E_{bonds}$ in the base simulation (330 T mol yr$^{-1}$).**





**4.2.2 Impact of changing isoprene emission on $E_{bonds}$**

As isoprene is the second largest source of oxidisable bonds (Fig. 3), increasing the isoprene emission results in a significant increase in $E_{bonds}$. Differences in both the spatial distribution of emissions and the oxidation chemistry of isoprene and $CH_4$, however, means that the impact of the increases in $E_{bonds}$ on $O_3$ production are significantly different for the two compounds. This is predominantly because the fraction of oxidisable bonds that are physically deposited for isoprene is high compared to those emitted as $CH_4$. This increase is due to i) the higher solubility of isoprene oxidation products compared to those of $CH_4$, and ii) the higher reactivity of isoprene means its oxidation occurs in the boundary layer where both dry and wet deposition is most effective.

Figure 10 shows the split of oxidisable bond loss mechanisms in the base simulation and those with the isoprene emissions multiplied by 0.5 and 2. The complex myriad of products formed during the isoprene oxidation mechanism also results in the production of many highly oxygenated multifunctional compounds with high Henrys law solubility constants, meaning they are more readily lost to deposition.

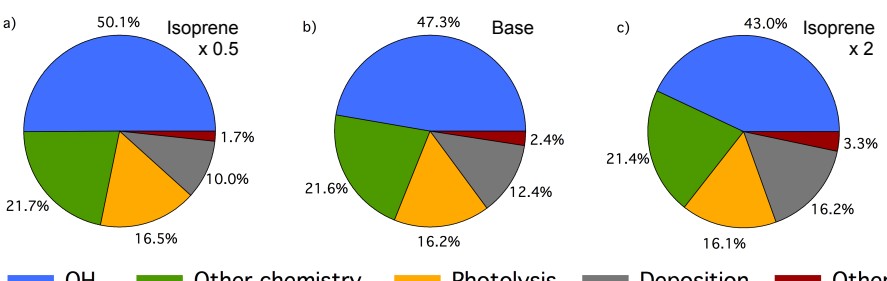

**Figure 10. Oxidisable bond loss mechanisms under changing isoprene emissions. Pie charts showing fractional loss mechanisms for oxidisable bonds in model simulations with 0.5 x isoprene emission (a), base simulation (b) and 2 x isoprene emission (c).**

Increasing the isoprene emission also has a slight offsetting impact on the effective $CH_4$ emission, as increased isoprene concentrations decrease OH concentrations, and thus decrease the effective $CH_4$ emission. A doubling in isoprene emission causes a 6% reduction in the effective emission of $CH_4$.





**4.3.3 Impact of changing isoprene emission on $F_{radicals}$, $F_{RO2}$ and $I$**
As shown in Fig. 3c(ii), increasing the isoprene emission results in a reduction in all
$P_sO_3$ efficiency parameters. The reductions in $F_{radicals}$ is due to the higher fraction of
oxidisable bonds that are lost via non-radical forming pathways (e.g. deposition) for
isoprene relative to the other main oxidisable bond emission sources $CH_4$ and CO.
The slight decreases of $F_{RO2}$ and $I$ with increasing isoprene emission are
predominantly due to changes in OH and $NO_x$ (Fig. 9).
The complex chemistry of isoprene oxidation combined with the spatial distribution of
isoprene emissions means the increase in $O_3$ production due to increases in isoprene
emissions is roughly half what might be expected from the increase in oxidisable bond
emission alone (i.e. if the increase was *via* CO instead of isoprene).
# 5. Conclusions
We have shown that this bond-focussed approach to $O_3$ production provides a
significantly more detailed understanding of the processes involved. The role of
modelled VOC emissions and $O_3$ burden has been reported previously [*Wild*, 2007;
*Young et al.*, 2013]. However previous efforts extending this to a general process led
approach has not been successful. This new approach provides a tool with which the
processes controlling $O_3$ production can be investigated, and a metric by which
different emissions can be compared. For example, the differing chemistry of isoprene
and $CH_4$ shows that even though their emissions of carbon mass are comparable, the
atmosphere responds in different ways, with the isoprene bonds being less effective in
producing $O_3$ than $CH_4$ bonds. By quantifying multiple steps in the $O_3$ production
process, competing changes in the system become apparent (as shown in Fig. 4b(ii)
and c(ii)) and are thus testable. This enables the effect of model approximations on $O_3$
production to be quantified (e.g. the effect of $NO_x$ on $CH_4$ emissions when using $CH_4$
concentration fields).
This new diagnostic also points towards the importance of observational datasets for
assessing our understanding of tropospheric chemistry. Although the budget presented
in Fig. 2 provides an annually integrated global estimate it points towards local
comparisons that can be made to assess model fidelity. Comparisons, both their
magnitude and their ratios, between observed and modelled bond concentration, bond
emission and loss fluxes (e.g. OH reactivity [*Yang et al.*, 2016] or depositional fluxes





[*Wesely and Hicks*, 2000]), and $O_3$ production [*Cazorla and Brune*, 2010] would all
provide comparisons for outputs from the $P_sO_3$ diagnostic and help assess model
performance.
Another potentially important application is in model-model comparisons. Increases
in our understanding of why different models calculate different $O_3$ production and
burdens has been slow [*Stevenson et al.*, 2006; *Wu et al.*, 2007; *Young et al.*, 2013]. A
comparison between models based on this methodology may well help identify at a
process level why models differ in their $O_3$ production. The application of this
diagnostic to regional $O_3$ production should also increase insight into the processes
controlling model $O_3$.



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

**Author contributions**
All work presented here was conceived by P.M.E. and M.J.E. The implementation,
model simulations and analysis were carried out by P.M.E., and the manuscript was
written by P.M.E. with substantial input from M.J.E..
**Additional information**
The authors declare no competing financial interests.
**Acknowledgements**
P.M.E. was supported by NERC Grant  NE/K004603/1. This work was also supported
by the NERC funded BACCHUS project (NE/L01291X/1).   GEOS-Chem
(www.geos-chem.org) is a community effort and we wish to thank all involved in the
development of the model.