# Peer review of "Manuscript under review for journal Atmos. Chem. Phys."

_Atmospheric Chemistry and Physics, 2017_

## Referee Comment (RC1) · Anonymous Referee #2 · 26 Jun 2017

*General Comments*

This paper presents a new method for diagnosing ozone production based on the processing of chemical bonds. The authors show that this new diagnostic changes our view of the relative importance of different hydrocarbon emissions, which is an improvement over previous methods using a simple total carbon-based approach. The authors also quantify the ozone-producing efficiency of the emitted bonds. The ability of this diagnostic to separate the difference between shifting the NO/NO$_2$ ratio and its impact on ozone production vs. the increase in the fraction of RO$_2$ reacting with NO is valuable. Overall, the discussion of the diagnostic and model sensitivities is quite lengthy and could be shortened by spending less time on the discussion of methane, per the comment below. This paper should be published after addressing the com-

ments below, in particular, how this diagnostic could be relevant to our understanding of the differences in ozone production across models without actually implementing the diagnostic in every single chemical transport model.

*Specific Comments*

The discussion of methane and isoprene is confusing due to the model implementation of methane as a fixed concentration. It might be better to focus the discussion on evaluating perturbations to isoprene emissions, and contrast that to methane, as opposed to the way it is presented now, with the caveat about model treatment of methane. Then the discussion of the dependence of methane 'emission' on OH would not be needed (i.e. Figure 7) which is difficult to follow.

This analysis would also be strengthened by presenting the types of information that global model comparisons of ozone production should include to take advantage of this type of diagnostic. For example, it seems that if all models presented their total methane, isoprene, CO, and $NO_x$ budgets, this diagnostic would help interpret the resulting impact on ozone production without actually implementing the diagnostic in each model. This might increase the scientific contribution of this paper.

The paragraph starting on line 341 needs clarification. What do you mean by "the final 20% due to the increased OH competing for the available oxidisable bonds." Doesn't this just mean that with higher $NO_x$, you get higher OH concentrations and thus you increase the concentration of $RO_2$ as well and NO?

*Technical Corrections*

Is discussing $SO_2$ oxidation relevant to ozone in any way? If not, it is confusing and should be removed.

You say that over a long enough timescale, the global atmosphere can be considered to be in steady-state, and thus equation (1) applies. Please clarify the conditions where this diagnostic is useful/applicable. For example, could it be used for a daily analysis

of ozone production.

Please be consistent with the use of $CH_3O_2$ or $MO_2$.

On line 438, the sentence that starts with "With the majority" is not a full sentence.

On line 440, remove the comma after OH.

———————————————————

---

## Referee Comment (RC2) · Anonymous Referee #1 · 20 Jul 2017

This paper presents a novel analysis of ozone production in terms on the spin states of the bonds in the precursor species. This is an interesting and original concept, and is a commendable attempt to generate a diagnostic of ozone production that has a sound physico-chemical basis, and one that provides more process insight than the standard methods based on NOx cycling. The paper is worthy of publication, but needs revision to address a number of weaknesses and to enhance its value to the scientific community.

General Comments

The background theory behind the diagnostic could be presented more clearly. While the concept of electron spin is well understood in the physical chemistry community, it is necessary to provide a brief introduction for a wider audience, along with references

to literature where readers can learn more.

The paper addresses the rate of ozone production, but discussion focuses solely on long-term integrated ozone production on an annual global scale. It is not clear how applicable the new diagnostic is to smaller regions and shorter timescales where the assumption of steady state (line 141) may be less appropriate, and where emissions may be less important than transport. What is needed to extend the diagnostic to these smaller spatial and temporal scales? The potential for analysis of regional budgets is alluded to on line 307, but no detail is provided.

The strengths and limitations of the approach should be set out more clearly. What additional insight does the new metric provide and how might this be applied to real problems (e.g., to the sensitivity of ozone production to assumptions of VOC speciation, to simplification of isoprene chemistry, or to treatments of deposition processes?) How does the approach compare with previous attempts to generate diagnostics, e.g., though the concept of photochemical ozone creation potentials (POCPs) for individual VOCs? There is little reference to earlier approaches in the field.

Specific Comments

Figure 4: please explain how the contributions of the R and F terms presented in the figure are derived. It is easy to see for the standard diagnostic, where the terms sum linearly to the total PO3, but it is not as clear for the new diagnostic as the terms are no longer independent of each other (as defined in Eq. 1).

The meaning of the horizontal dashed lines in Figure 4 is not clear.

l.270: How many simulations were performed for these sensitivity studies? Please state this in the text.

Figure 7 is not well conceived. It is not clear why a log-NOx scale is used, given that the relationships expected are not exponential (neither line drawn here is expected to be straight, as would quickly become evident at larger or smaller NOx levels). Perhaps

plot OH vs CH4 bond emission directly, and label the points with the NOx level?

Figures 8 and 10 would be more effectively presented through the use of a bar chart, so that the relative changes can be seen more clearly.

Supplement: The "errors in chemistry scheme" need some explanation, and these entries should be at the bottom of the table, as it doesn't aid the reader's comprehension to put them at the top.

The supplement needs more detail on the implementation of the approach. It would be difficult for anyone to replicate in a different model without more information about the reaction classification. It would be helpful to provide a worked example of how the multiple in the Table is arrived at, and this could be included in the supplement.

Typos and minor issues

The English grammar needs a little work in places, particularly where the subject of a verb is inappropriate (e.g., "GEOS-Chem fixes CH4 concentrations..." on line 318 would be clearer as "CH4 concentrations are fixed in GEOS-Chem...")

l.133: add the before top

l.251: grammar in first sentence needs correcting.

l.358: MO2 should be written as CH3O2 for consistency with line 356, and it would be helpful to do this throughout the text, e.g., line 427/8.

Numbers less than 10 without units are better presented as text than numerals.

—————————————————

---

## Author Comment (AC1) · 4 Sep 2017

Anonymous Referee #1 This paper presents a novel analysis of ozone production in terms on the spin states of the bonds in the precursor species. This is an interesting and original concept, and is a commendable attempt to generate a diagnostic of ozone production that has a sound physico-chemical basis, and one that provides more process insight than the standard methods based on NOx cycling. The paper is worthy of publication, but needs revision to address a number of weaknesses and to enhance its value to the scientific community.

General Comments

1) The background theory behind the diagnostic could be presented more clearly.

[Figure]

While the concept of electron spin is well understood in the physical chemistry community, it is necessary to provide a brief introduction for a wider audience, along with references to literature where readers can learn more.

Response: The paragraph in the introduction that introduces spin has been expanded and a reference to an explanation of the fundamental principles included. This paragraph now reads:

Change to manuscript: The inefficiency of ground state O2 as an atmospheric oxidant is due to its electronic structure. In quantum mechanics, all atomic particles have an intrinsic angular momentum known as spin [Atkins and De Paula, 2014]. The spin of an electron is described by the spin quantum number, s, and can have values of either $+\frac{1}{2}$ or $-\frac{1}{2}$ for a single electron. The Pauli exclusion principle states that if two electrons occupy the same orbital then their spins must be paired, and thus cancel. With two unpaired electrons ground state O2 is a spin-triplet with a total spin quantum number $S=\frac{1}{2}+\frac{1}{2}=1$ (giving a term symbol of ($_{}^3)\Sigma_g^-$ ). In contrast, virtually all trace chemicals emitted into the atmosphere contain only paired electrons and are thus spin-singlets (S=0). The quantum mechanical spin selection rule $\Delta$S=0 means that allowed electronic transitions must not result in a change in electron spin. From a simplistic perspective (i.e. ignoring nuclear spin interactions, inter-system crossings, nuclear dipole effects etc.) this spin selection rule means that the reaction of ground state O2 with most emitted compounds is effectively spin forbidden. Electronically excited O2 (($_{}^1)\Delta_g^{}$ or ($_{}^1)\Sigma_g^+$ ) is a spin singlet and is more reactive in the atmosphere but low concentrations limit its role [Larson and Marley, 1999]. Instead, atmospheric oxidation proceeds predominantly via reactions with spin-doublet oxygen-derived species ($S=\frac{1}{2}$), notably the hydroxyl (OH) and peroxy radicals (RO2 = HO2, CH3O2, C2H5O2, etc.), or spin-singlet species (e.g. ozone (O3)).

2) The paper addresses the rate of ozone production, but discussion focuses solely on long-term integrated ozone production on an annual global scale. It is not clear how applicable the new diagnostic is to smaller regions and shorter timescales where the

assumption of steady state (line 141) may be less appropriate, and where emissions may be less important than transport. What is needed to extend the diagnostic to these smaller spatial and temporal scales? The potential for analysis of regional budgets is alluded to on line 307, but no detail is provided.

The strengths and limitations of the approach should be set out more clearly. What additional insight does the new metric provide and how might this be applied to real problems (e.g., to the sensitivity of ozone production to assumptions of VOC speciation, to simplification of isoprene chemistry, or to treatments of deposition processes?) How does the approach compare with previous attempts to generate diagnostics, e.g., though the concept of photochemical ozone creation potentials (POCPs) for individual VOCs? There is little reference to earlier approaches in the field.

Response: The aim of this paper is to describe a new approach for the study of ozone production in chemical transport models, and to illustrate this through a global budget analysis and comparison with the most commonly used diagnostic for this. The application of the new diagnostic to other scales and problems, as well as comparison to other available metrics is of interest but unfortunately outside the scope of this work. The following paragraph has been added to the conclusion section to discuss the strengths of the approach in relation to other possible applications, and also identify things that would need to be considered for this to be successful.

Change to manuscript: Future work is necessary to identify the usefulness of this approach on smaller spatial and temporal scales. For regional modelling scale, the transport flux of bonds into the domain would need to be considered alongside the emissions of bonds. However, this might help to disentangle O3 production due to local VOC emissions from that due to VOC emissions outside of the domain. This bond focussed approach may also have usefulness on shorter timescales. For example, when considering vertical fluxes in and out of the boundary layer, a bond centred approach could help. What fraction of the bonds emitted at the surface are exported to the free troposphere. If a measurement of reactivity flux could be made this could be

tested experimentally.

Specific Comments

3) Figure 4: please explain how the contributions of the R and F terms presented in the figure are derived. It is easy to see for the standard diagnostic, where the terms sum linearly to the total PO3, but it is not as clear for the new diagnostic as the terms are no longer independent of each other (as defined in Eq. 1).

Response: The following text has been added to the figure caption.

Change to manuscript: The PsO3 diagnostic parameters are derived for each model simulation using the diagnostic implementation described in Sect. 3, and the fractional change in each parameter from the base simulation calculated.

4) The meaning of the horizontal dashed lines in Figure 4 is not clear.

Response: These are gridlines to aid comparison between plots. We have not changed this as we feel it helps interpretation of the figure, but are happy to take the editors guidance.

5) l.270: How many simulations were performed for these sensitivity studies? Please state this in the text.

Response: Text now includes following sentence.

Change to manuscript: A set of 5 simulations was performed for each model sensitivity investigated (NOx, isoprene and CH4), with a common base simulation, resulting in 13 simulations in total.

6) Figure 7 is not well conceived. It is not clear why a log-NOx scale is used, given that the relationships expected are not exponential (neither line drawn here is expected to be straight, as would quickly become evident at larger or smaller NOx levels). Perhaps plot OH vs CH4 bond emission directly, and label the points with the NOx level?

Response: We thank the reviewer for this suggestion and have updated the figure to that attached (attached Fig1).

Change to manuscript: Updated figure with new caption "Figure 7. Effective CH4 emissions as a function of global mean OH concentration, for simulations where NOx emissions were changed. Marker size and colour indicate global NOx concentration.".

7) Figures 8 and 10 would be more effectively presented through the use of a bar chart, so that the relative changes can be seen more clearly.

Response: We again thank the reviewer for this suggestion and have remade the figures as bar charts (see attached Figs 2 & 3).

Change to manuscript: Updated figures with new captions "Figure 8. Oxidisable bond loss mechanism fractions under changing effective CH4 emissions (0.5 x CH4 concentration field, base simulation and 2 x CH4 concentration field)." and "Figure 10. Oxidisable bond loss mechanism fractions under changing isoprene emissions.".

8) Supplement: The "errors in chemistry scheme" need some explanation, and these entries should be at the bottom of the table, as it doesn't aid the reader's comprehension to put them at the top.

Response: We have added the following text to the supplement and moved the entries to the bottom of table S1.

Change to manuscript: Inconsistencies within the chemistry scheme, where the lumped nature of some reactions result in a non-physical production or loss of oxidisable bonds, are also tracked as errors in the chemistry scheme.

9) The supplement needs more detail on the implementation of the approach. It would be difficult for anyone to replicate in a different model without more information about the reaction classification. It would be helpful to provide a worked example of how the multiple in the Table is arrived at, and this could be included in the supplement.

Response: The following has been added to the supplement.

Change to manuscript: Reaction tags were added to all reactions in the chemistry scheme, and the GEOS-Chem diagnostic was used to provide a direct measure of their production. An example of how this was implemented is shown below for a select few steps of the methane oxidation scheme illustrated in Fig. 1. CH4 + OH → CH3O2 + 1[Tag1] + 1[Tag2] + 1[Tag3] CH3O2 + HO2 → CH3OOH + 1[Tag4] + 1[Tag5] CH3O2 + CH3O2 → CH3OH + CH2O + 2[Tag6] + 1[Tag7] Reaction tags used in example reactions: Tag1 = Oxidisable bond lost via OH chemical reaction; Tag2 = Oxidisable bond + OH → 1 radical (RO2); Tag3 = OH + CH4 reaction (special tag used to calculate effective CH4 emission); Tag4 = RO2 to peroxide; Tag5 = HO2 to peroxide; Tag6 = RO2 lost to carbonyl forming peroxy radical self reaction; Tag7 = Bond lost to RO2 + RO2 → 0 radicals. Typos and minor issues

10) The English grammar needs a little work in places, particularly where the subject of a verb is inappropriate (e.g., "GEOS-Chem fixes CH4 concentrations..." on line 318 would be clearer as "CH4 concentrations are fixed in GEOS-Chem...")

Response: This has been addressed.

11) l.133: add the before top

Response: This has been addressed.

12) l.251: grammar in first sentence needs correcting.

Response: This has been addressed.

13) l.358: MO2 should be written as CH3O2 for consistency with line 356, and it would be helpful to do this throughout the text, e.g., line 427/8.

Response: This has been addressed.

14) Numbers less than 10 without units are better presented as text than numerals.

Response: This stylistic change has not been implemented, but the authors are happy to do so if the editor wishes.

**Fig. 1.** Figure 7. Effective CH4 emissions as a function of global mean OH concentration, for simulations where NOx emissions were changed. Marker size and colour indicate global NOx concentration.

[Figure]

**Fig. 2.** Figure 8. Oxidisable bond loss mechanism fractions under changing effective CH4 emissions (0.5 x CH4 concentration field, base simulation and 2 x CH4 concentration field).

[Figure]

**Fig. 3.** Figure 10. Oxidisable bond loss mechanism fractions under changing isoprene emissions.

---

## Author Comment (AC2) · 4 Sep 2017

General Comments This paper presents a new method for diagnosing ozone production based on the processing of chemical bonds. The authors show that this new diagnostic changes our view of the relative importance of different hydrocarbon emissions, which is an improvement over previous methods using a simple total carbon-based approach. The authors also quantify the ozone-producing efficiency of the emitted bonds. The ability of this diagnostic to separate the difference between shifting the NO/NO2 ratio and its impact on ozone production vs. the increase in the fraction of RO2 reacting with NO is valuable. Overall, the discussion of the diagnostic and model sensitivities

is quite lengthy and could be shortened by spending less time on the discussion of methane, per the comment below. This paper should be published after addressing the comments below, in particular, how this diagnostic could be relevant to our understanding of the differences in ozone production across models without actually implementing the diagnostic in every single chemical transport model.

Specific Comments

1) The discussion of methane and isoprene is confusing due to the model implementation of methane as a fixed concentration. It might be better to focus the discussion on evaluating perturbations to isoprene emissions, and contrast that to methane, as opposed to the way it is presented now, with the caveat about model treatment of methane. Then the discussion of the dependence of methane 'emission' on OH would not be needed (i.e. Figure 7) which is difficult to follow.

Response: We accept the reviewers comment that the discussion of methane and isoprene could be confusing. However, the fundamental differences in both their chemistries and treatment in the majority of chemical transport models mean we strongly feel that they warrant individual treatment. We have significantly shortened and simplified the discussion of methane, and have simplified Fig. 7 (see response to referee #1 comment 6). As the 1st reviewer did not have an issue with the individual discussions of methane and isoprene we respectfully leave it to the editor to decide if our response to this comment is adequate.

2) This analysis would also be strengthened by presenting the types of information that global model comparisons of ozone production should include to take advantage of this type of diagnostic. For example, it seems that if all models presented their total methane, isoprene, CO, and NOx budgets, this diagnostic would help interpret the resulting impact on ozone production without actually implementing the diagnostic in each model. This might increase the scientific contribution of this paper.

Response: We have added the following paragraph to the conclusions section of the

paper.

Changes to manuscript: Another potentially important application is in model-model comparisons. Increases in our understanding of why different models calculate different O3 production and burdens has been slow [Stevenson et al., 2006; Wu et al., 2007; Young et al., 2013]. Although a complete tagging like that described here is unlikely to occur for all of the models involved in the comparison, a small number of additional diagnostics is likely to produce a significantly better understanding of the models. Diagnosing (1) the total bond flux (direct emissions plus the flux for those species kept constant), (2) the rate of production of RO2 and (3) the rate of production of O3, could help differentiate why certain models produce more or less O3 than others. The ratios between these fluxes would help identify what aspect of the emissions of chemistry differs between the models.

3) The paragraph starting on line 341 needs clarification. What do you mean by "the final 20% due to the increased OH competing for the available oxidisable bonds." Doesn't this just mean that with higher NOx, you get higher OH concentrations and thus you increase the concentration of RO2 as well and NO?

Response: This sentence has been reworded to avoid confusion.

Changes to manuscript: "the final 20% due to the increased OH concentration competing for the available oxidisable bonds and resulting in increased RO2 production."

Technical Corrections

4) Is discussing SO2 oxidation relevant to ozone in any way? If not, it is confusing and should be removed.

Response: Although SO2 oxidation has minimal direct impact on O3 production it is still a source of peroxy radicals that are central to this diagnostic approach (SO2 + OH + O2 → SO3 + HO2). We therefore would prefer to keep the sentence on SO2 for completeness and also to aid others in reproducing the diagnostic approach.

5) You say that over a long enough timescale, the global atmosphere can be considered to be in steady-state, and thus equation (1) applies. Please clarify the conditions where this diagnostic is useful/applicable. For example, could it be used for a daily analysis of ozone production.

Response See response to referee #1 comment 2.

6) Please be consistent with the use of CH3O2 or MO2.

Response: This has been addressed.

7) On line 438, the sentence that starts with "With the majority" is not a full sentence.

Response: This has been addressed.

8) On line 440, remove the comma after OH.

Response: This has been addressed.

---

## Author Response (AR1)

Response to reviewers comments for:
"A new diagnostic for tropospheric ozone production"
(acp-2017-378)

The reviewers comments have been addressed. Below are the authors responses (red) and changes implemented (blue). The reviewers comments (black) have been numbered to avoid confusion.

**Anonymous Referee #1**
This paper presents a novel analysis of ozone production in terms on the spin states of the bonds in the precursor species. This is an interesting and original concept, and is a commendable attempt to generate a diagnostic of ozone production that has a sound physico-chemical basis, and one that provides more process insight than the standard methods based on NOx cycling. The paper is worthy of publication, but needs revision to address a number of weaknesses and to enhance its value to the scientific community.

General Comments

1. The background theory behind the diagnostic could be presented more clearly. While the concept of electron spin is well understood in the physical chemistry community, it is necessary to provide a brief introduction for a wider audience, along with references to literature where readers can learn more.

   The paragraph in the introduction that introduces spin has been expanded and a reference to an explanation of the fundamental principles included. This paragraph now reads:

   The inefficiency of ground state $O_2$ as an atmospheric oxidant is due to its electronic structure. In quantum mechanics, all atomic particles have an intrinsic angular momentum known as spin [*Atkins and De Paula*, 2014]. The spin of an electron is described by the spin quantum number, s, and can have values of either +½ or -½ for a single electron. The Pauli exclusion principle states that if two electrons occupy the same orbital then their spins must be paired, and thus cancel. With two unpaired electrons ground state $O_2$ is a spin-triplet with a total spin quantum number S=½+½=1 (giving a term symbol of $^3\Sigma_g^-$). In contrast, virtually all trace chemicals emitted into the atmosphere contain only paired electrons and are thus spin-singlets (S=0). The quantum mechanical spin selection rule ΔS=0 means that allowed electronic transitions must not result in a change in electron spin. From a simplistic perspective (i.e. ignoring nuclear spin interactions, inter-system crossings, nuclear dipole effects etc.) this spin selection rule means that the reaction of ground state $O_2$ with most emitted compounds is effectively spin forbidden. Electronically excited $O_2$ ($^1\Delta_g$ or $^1\Sigma_g^+$) is a spin singlet and is more reactive in the atmosphere but low concentrations limit its role [*Larson and Marley*, 1999]. Instead, atmospheric oxidation proceeds predominantly via reactions with spin-doublet oxygen-derived species (S=½), notably the hydroxyl (OH) and peroxy radicals ($RO_2 = HO_2$, $CH_3O_2$, $C_2H_5O_2$, etc.), or spin-singlet species (e.g. ozone ($O_3$)).

2. The paper addresses the rate of ozone production, but discussion focuses solely on long-term integrated ozone production on an annual global scale. It is not clear how applicable the new diagnostic is to smaller regions and shorter timescales where the assumption of steady state (line 141) may be less appropriate, and where emissions may be less important than transport. What is needed to extend the diagnostic to these smaller spatial and temporal scales? The potential for analysis of regional budgets is alluded to on line 307, but no detail is provided.

The strengths and limitations of the approach should be set out more clearly. What additional insight does the new metric provide and how might this be applied to real problems (e.g., to the sensitivity of ozone production to assumptions of VOC speciation, to simplification of isoprene chemistry, or to treatments of deposition processes?) How does the approach compare with previous attempts to generate diagnostics, e.g., though the concept of photochemical ozone creation potentials (POCPs) for individual VOCs? There is little reference to earlier approaches in the field.

The aim of this paper is to describe a new approach for the study of ozone production in chemical transport models, and to illustrate this through a global budget analysis and comparison with the most commonly used diagnostic for this. The application of the new diagnostic to other scales and problems, as well as comparison to other available metrics is of interest but unfortunately outside the scope of this work. The following paragraph has been added to the conclusion section to discuss the strengths of the approach in relation to other possible applications, and also identify things that would need to be considered for this to be successful.

Future work is necessary to identify the usefulness of this approach on smaller spatial and temporal scales. For regional modelling scale, the transport flux of bonds into the domain would need to be considered alongside the emissions of bonds. However, this might help to disentangle $O_3$ production due to local VOC emissions from that due to VOC emissions outside of the domain. This bond focussed approach may also have usefulness on shorter timescales. For example, when considering vertical fluxes in and out of the boundary layer, a bond centred approach could help. What fraction of the bonds emitted at the surface are exported to the free troposphere. If a measurement of reactivity flux could be made this could be tested experimentally.

Specific Comments

3. Figure 4: please explain how the contributions of the R and F terms presented in the figure are derived. It is easy to see for the standard diagnostic, where the terms sum linearly to the total PO3, but it is not as clear for the new diagnostic as the terms are no longer independent of each other (as defined in Eq. 1).

    The following text has been added to the figure caption.

    The $P_sO_3$ diagnostic parameters are derived for each model simulation using the diagnostic implementation described in Sect. 3, and the fractional change in each parameter from the base simulation calculated.

4. The meaning of the horizontal dashed lines in Figure 4 is not clear.

    These are gridlines to aid comparison between plots. We have not changed this as we feel it helps interpretation of the figure, but are happy to take the editors guidance.

5. l.270: How many simulations were performed for these sensitivity studies? Please state this in the text.

    Text now includes following sentence.

A set of 5 simulations was performed for each model sensitivity investigated ($NO_x$, isoprene and $CH_4$), with a common base simulation, resulting in 13 simulations in total.

6. Figure 7 is not well conceived. It is not clear why a log-NOx scale is used, given that the relationships expected are not exponential (neither line drawn here is expected to be straight, as would quickly become evident at larger or smaller NOx levels). Perhaps plot OH vs CH4 bond emission directly, and label the points with the NOx level?

We thank the reviewer for this suggestion and have updated the figure to that shown below.

[Figure]

Figure 7. Effective $CH_4$ emissions as a function of global mean OH concentration, for simulations where $NO_x$ emissions were changed. Marker size and colour indicate global $NO_x$ concentration.

7. Figures 8 and 10 would be more effectively presented through the use of a bar chart, so that the relative changes can be seen more clearly.

We again thank the reviewer for this suggestion and have remade the figures as bar charts (see below).

[Figure]

**Figure 8. Oxidisable bond loss mechanism fractions under changing effective CH₄ emissions (0.5 x CH₄ concentration field, base simulation and 2 x CH₄ concentration field).**

[Figure]

**Figure 10. Oxidisable bond loss mechanism fractions under changing isoprene emissions.**

8.  Supplement: The "errors in chemistry scheme" need some explanation, and these entries should be at the bottom of the table, as it doesn't aid the reader's comprehension to put them at the top.

We have added the following text to the supplement and moved the entries to the bottom of table S1.

Inconsistencies within the chemistry scheme, where the lumped nature of some reactions result in a non-physical production or loss of oxidisable bonds, are also tracked as errors in the chemistry scheme.

9. The supplement needs more detail on the implementation of the approach. It would be difficult for anyone to replicate in a different model without more information about the reaction classification. It would be helpful to provide a worked example of how the multiple in the Table is arrived at, and this could be included in the supplement.

The following has been added to the supplement.

Reaction tags were added to all reactions in the chemistry scheme, and the GEOS-Chem diagnostic was used to provide a direct measure of their production. An example of how this was implemented is shown below for a select few steps of the methane oxidation scheme illustrated in Fig. 1.

$CH_4 + OH \rightarrow CH_3O_2 + 1[Tag1] + 1[Tag2] + 1[Tag3]$

$CH_3O_2 + HO_2 \rightarrow CH_3OOH + 1[Tag4] + 1[Tag5]$

$CH_3O_2 + CH_3O_2 \rightarrow CH_3OH + CH_2O + 2[Tag6] + 1[Tag7]$

Reaction tags used in example reactions: Tag1 = Oxidisable bond lost via OH chemical reaction; Tag2 = Oxidisable bond + OH $\rightarrow$ 1 radical ($RO_2$); Tag3 = OH + $CH_4$ reaction (special tag used to calculate effective $CH_4$ emission); Tag4 = $RO_2$ to peroxide; Tag5 = $HO_2$ to peroxide; Tag6 = $RO_2$ lost to carbonyl forming peroxy radical self reaction; Tag7 = Bond lost to $RO_2 + RO_2 \rightarrow 0$ radicals.

Typos and minor issues

10. The English grammar needs a little work in places, particularly where the subject of a verb is inappropriate (e.g., "GEOS-Chem fixes CH4 concentrations..." on line 318 would be clearer as "CH4 concentrations are fixed in GEOS-Chem...")

This has been addressed.

11. l.133: add the before top

This has been addressed.

12. l.251: grammar in first sentence needs correcting.

This has been addressed.

13. l.358: MO2 should be written as CH3O2 for consistency with line 356, and it would be helpful to do this throughout the text, e.g., line 427/8.

This has been addressed.

14. Numbers less than 10 without units are better presented as text than numerals.

This stylistic change has not been implemented, but the authors are happy to do so if the editor wishes.

**Anonymous Referee #2**

*General Comments*

This paper presents a new method for diagnosing ozone production based on the processing of chemical bonds. The authors show that this new diagnostic changes our view of the relative importance of different hydrocarbon emissions, which is an improvement over previous methods using a simple total carbon-based approach. The authors also quantify the ozone-producing efficiency of the emitted bonds. The ability of this diagnostic to separate the difference between shifting the NO/NO2 ratio and its impact on ozone production vs. the increase in the fraction of RO2 reacting with NO is valuable. Overall, the discussion of the diagnostic and model sensitivities is quite lengthy and could be shortened by spending less time on the discussion of methane, per the comment below. This paper should be published after addressing the comments below, in particular, how this diagnostic could be relevant to our understanding of the differences in ozone production across models without actually implementing the diagnostic in every single chemical transport model.

*Specific Comments*

15. The discussion of methane and isoprene is confusing due to the model implementation of methane as a fixed concentration. It might be better to focus the discussion on evaluating perturbations to isoprene emissions, and contrast that to methane, as opposed to the way it is presented now, with the caveat about model treatment of methane. Then the discussion of the dependence of methane 'emission' on OH would not be needed (i.e. Figure 7) which is difficult to follow.

We accept the reviewers comment that the discussion of methane and isoprene could be confusing. However, the fundamental differences in both their chemistries and treatment in the majority of chemical transport models mean we strongly feel that they warrant individual treatment. We have significantly shortened and simplified the discussion of methane, and have simplified Fig. 7 (see comment 6 above). As the 1$^{st}$ reviewer did not have an issue with the individual discussions of methane and isoprene we respectfully leave it to the editor to decide if our response to this comment is adequate.

16. This analysis would also be strengthened by presenting the types of information that global model comparisons of ozone production should include to take advantage of this type of diagnostic. For example, it seems that if all models presented their total methane, isoprene, CO, and NOx budgets, this diagnostic would help interpret the resulting impact on ozone production without actually implementing the diagnostic in each model. This might increase the scientific contribution of this paper.

We have added the following paragraph to the conclusions section of the paper.

Another potentially important application is in model-model comparisons. Increases in our understanding of why different models calculate different $O_3$ production and burdens has been slow [*Stevenson et al.*, 2006; *Wu et al.*, 2007; *Young et al.*, 2013]. Although a complete tagging like that described here is unlikely to occur for all of the models involved in the comparison, a small number of additional diagnostics is likely to produce a significantly better understanding of the models. Diagnosing (1) the total bond flux (direct emissions plus the flux for those species kept constant), (2) the rate of production of $RO_2$ and (3) the rate of production of $O_3$, could help differentiate why certain models produce more or less $O_3$ than others. The ratios between these fluxes would help identify what aspect of the emissions of chemistry differs between the models.

17. The paragraph starting on line 341 needs clarification. What do you mean by "the final 20% due to the increased OH competing for the available oxidisable bonds." Doesn't this just mean that with higher NOx, you get higher OH concentrations and thus you increase the concentration of RO2 as well and NO?

This sentence has been reworded to avoid confusion.

"the final 20% due to the increased OH concentration competing for the available oxidisable bonds and resulting in increased $RO_2$ production."

*Technical Corrections*

18. Is discussing SO2 oxidation relevant to ozone in any way? If not, it is confusing and should be removed.

Although $SO_2$ oxidation has minimal direct impact on $O_3$ production it is still a source of peroxy radicals that are central to this diagnostic approach ($SO_2 + OH + O_2 \rightarrow SO_3 + HO_2$). We therefore would prefer to keep the sentence on $SO_2$ for completeness and also to aid others in reproducing the diagnostic approach.

19. You say that over a long enough timescale, the global atmosphere can be considered to be in steady-state, and thus equation (1) applies. Please clarify the conditions where this diagnostic is useful/applicable. For example, could it be used for a daily analysis of ozone production.

See response to comment 2.

20. Please be consistent with the use of CH3O2 or MO2.

This has been addressed.

21. On line 438, the sentence that starts with "With the majority" is not a full sentence.

This has been addressed.

22. On line 440, remove the comma after OH.

This has been addressed.

**A new diagnostic for tropospheric ozone production**

Peter M. Edwards[1]* & Mathew J. Evans[1,2]

[revised manuscript text omitted]